# The Association between Working Memory and Divergent Thinking: The Moderating Role of Formal Musical Background

**DOI:** 10.3390/brainsci14010061

**Published:** 2024-01-08

**Authors:** Maria Chiara Pino, Marco Giancola, Massimiliano Palmiero, Simonetta D’Amico

**Affiliations:** 1Department of Biotechnology and Applied Clinical Sciences, University of L’Aquila, 67100 L’Aquila, Italy; marco.giancola@univaq.it (M.G.); simonetta.damico@univaq.it (S.D.); 2Department of Communication Sciences, University of Teramo, 64100 Teramo, Italy; mpalmiero@unite.it

**Keywords:** formal musical background, working memory, divergent thinking, youth, moderation

## Abstract

Divergent thinking (DT) is widely considered an essential cognitive dimension of creativity, which involves goal-oriented processes, including working memory (WM), which allows for retrieving and loading of information into the attentional stream and, consequently, enhancing divergence of thinking. Despite the critical role of WM in DT, little work has been done on the mechanism affecting this interplay. The current study addressed the involvement of a formal musical background in the relationship between WM and DT and was conducted with 83 healthy young adults (M = 19.64 years; SD = 0.52 years; 33 females). The participants were requested to indicate if they had a formal background in music in the conservatory (M = 4.78 years; SD = 5.50 years) as well as perform the digit span forward test (DSFT) and the alternative uses task—AUT from the Torrance test of creative thinking (TTCT). The results indicated that years of formal musical background moderated the association between WM and DT. These findings suggest that music enhances the positive effect of high-order cognitive processes, such as WM, on the ability to think divergently. Theoretical and practical implications as well as limitations were discussed.

## 1. Introduction

Creativity is a dynamic process that is necessary to solve a variety of problems, provides interesting solutions, and requires a combination of “some” originality and usefulness [1,2]. Creativity, along with critical thinking, communication, and collaboration, is explicitly identified as one of the four most important 21st-century skills and is strongly emphasized in school curricula around the world [3,4]. Creativity can also be defined as a form of survival, as it allows us to constantly adapt to new circumstances [5]. Recently, creativity has been widely recognized as a crucial factor for social and economic growth as well as solving problems in daily life [6]. One of the most important components of creativity is divergent thinking (DT henceforth), which reflects the ability to generate creative ideas by combining different types of information in novel ways [7]. DT is opposed to convergent thinking, which is oriented toward finding one solution to a clearly defined problem or situation [8], which is useful when a task requests an individual to give the single best or most often correct answer to a question. One of the strengths of DT is that it incorporates the ability to find alternative and new solutions to open-ended problems, break schemes, and be as creative as possible [9].

DT is not only an index of creative potential [10] but also a strategy for solving ill-defined problems [11]. Neuroimaging studies have shown that it is supported by simultaneously engaging the default, salience, and executive brain systems [12,13]. Executive functions, a set of cognitive and emotional skills that are the basis for goal-directed thinking and behavior, seem to play a key role in promoting DT. There are three ways in which executive functions can affect DT: (i) through Working Memory (WM) requesting holding in mind task instructions, keeping track of information, and mentally manipulating information to generate new ideas [14,15]; (ii) through cognitive flexibility that could break out of habitual models of thinking [16]; and (iii) through inhibitory control that involves the ability to control one’s attention, behavior, thoughts, and/or emotions, creating the possibility of change and choice, and it is necessary for overriding interference from obvious ideas [16,17].

Moreover, DT is influenced by many other factors due to its complexity and multidimensionality [4]. Among them, the key role of music practice can be considered. Music practice seems to have a strong effect on promoting cognitive and emotional skills. In particular, the musical background seems to facilitate the relationship between the ability to receive and process information (i.e., WM) and the ability to recombine information in new ways to generate new ideas to solve ill-defined problems (i.e., DT) based on the focused attention mechanism. Accordingly, to these premises, the current research aims to provide further evidence on the association between WM and DT, addressing the involvement of a formal musical background and proposing a moderation model.

### 1.1. Working Memory and Divergent Thinking

Previous research on the psychology of creativity has suggested that DT relies on a multitude of factors, including personality traits, such as the Big Five, Dark Triad, and Trait Emotional Intelligence [18,19,20], as well as controlled cognitive processes [21,22]. Among these cognitive processes, WM was found to play a critical role in DT (for a detailed review see [23]) given that it is involved in the maintenance and manipulation of information over short periods [24,25]. Indeed, to generate ideas, one must hold in mind task instructions, keep track of ideas to avoid repeating them, and mentally manipulate information; all these activities require WM ability. WM is a type of short-term memory involving the ability to hold and manipulate internal and external information and is fundamental to the organization of goal-directed behavior [26]. It is worth noting that working memory and short-term memory are often used to refer to the same skill, but these two systems have different properties: short-term memory only allows us to retain information, while WM is an active capacity to temporarily store and simultaneously process information [27]. This latter seems to be mostly related to the mechanisms underlying DT.

The role of WM in creative ideation is determined by a cognitive process that allows WM to consolidate information into a durable and persistent representation that can be retained in WM or manipulated to achieve the goals a person wants to accomplish [28]. When a stimulus first occurs, the initial sensory trace is frugal and susceptible to forgetfulness [29]. A model of WM consolidation supports the idea that it can stabilize representations by combining features of the stimulus, context, and associated prior knowledge [28,30]. This WM consolidation process allows creative ability to disassemble and recombine the old information contained in memory in new ways [17]. Thus, creating new ideas is not possible without involving working memory. In this way, the combined functioning of WM and DT enables us to relate multiple ideas or facts together by recombining them in new ways and resisting the repetition of old thought patterns [17]. If we take the relationship between WM and DT to be true, then more creative people should also have a better WM ability to consolidate [30]. Indeed, even though some studies have shown no specific relationship between WM and DT, high WM has been found to be relevant for DT under specific circumstances, such as enhanced persistence and consequently focused attention, or at specific stages of the idea-generation process. As further evidence of the association between WM and DT, working memory updating training has yielded reductions in activation in the ventrolateral and dorsolateral prefrontal cortex, two brain areas that play a critical role in DT [4]. Yet, previous studies showed that other processes can mediate the relationship between WM and DT, such as task instructions, intelligence, and associative fluency (see, [23]). This means that the extent to which high WM performance can positively affect DT, allowing the retrieving and loading of more information into the attentional stream and, consequently, increasing divergence [19], must be clarified considering other factors.

### 1.2. Formal Musical Background and Divergent Thinking

Musical practices have recently attracted the attention of research focusing on their creative properties and the creative potential of musicians [31]. Indeed, a typical cliché of musicians is that they are considered predominantly artistic individuals, meaning that they are creative and original. Practicing music is certainly an intense and multisensory experience that requires the acquisition and maintenance of a range of cognitive and motor skills throughout a musician’s life [32]. Indeed, music practice increases a wide range of cognitive abilities, such as visuospatial reasoning, processing speed, and WM [33], from the early stages of life [33,34,35]. For this reason, musicians are considered an excellent human model for the study of behavioral, cognitive, and brain effects in the acquisition, practice, maintenance, and integration of sensory, cognitive, and motor skills.

Scientific interest in the potential benefits of music on cognitive skills may stem from studies of the “Mozart effect,” a controversial scientific theory developed in 1993 by the physicists Gordon Shaw and Frances Rauscher, which suggested the cognitive abilities of people who listened to music by Mozart improved compared to those who performed their cognitive task without listening to this music [36] (in their first study, the Mozart effect was demonstrated for visuospatial skills). Following this research, Adaman and Blaney [37] published a study that used music to induce different moods (depressed, elated, or neutral mood), followed by a divergent creativity task where the participants had to generate alternative uses for a household item. The authors showed a significant effect regarding an increase in fluency, in terms of the number of uses generated, for both ‘depressed’ and ‘elated’ moods when compared with a ‘neutral’ mood [37].

Therefore, there could be a strong connection between musical practice and DT. It is no coincidence that the musician Gerald Klickstein defined the practice in the musical field as a “*deliberate, creative process of improving musical ability and mastering music for performance*”. Nevertheless, it is necessary to emphasize how music is based on statistical learning characterized by specific rules to be followed. Indeed, music can be defined as a hierarchical system in which smaller and separate units (notes) are combined into high-order structures by specific rules (musical compositions) [35,38].

Several studies have suggested that experience in the music field enhances DT [39,40,41,42] in terms of fluency, flexibility, and originality. Strengthening the associative modes of processing, which facilitate the retrieval of information from long-term memory, and improving the WM competences, which facilitate the online recombination of information [23], might explain the relationship between musical practice and DT. In other words, the effect that a musical background might have on DT calls into question both attention and mnemonic mechanisms, particularly WM. This scenario offers the grounds to assume that musical practice and WM jointly affect DT.

### 1.3. The Moderating Role of Formal Musical Background

Overall, considering the relationship between WM and DT, already extensively documented in the literature, and the key role that music practice seems to play in both these skills, we hypothesized that a formal musical background could facilitate this relationship. First, musical practice involves focused attention [33] as well as WM and DT. Musicians need a selective and flexible attention ability, as well as inhibition of nonsignificant auditory and visual stimuli [43]. Indeed, musical activity involves long periods of controlled attention, by retaining musical passages in WM or encoding them in long-term memory as well as decoding and transforming notes into corresponding motor commands [43]. Yet, WM relies on inhibition and, consequently, on focused attention [44], given that holding a specific type of information necessarily requires inhibiting other information. DT also requires focused attention: highly divergent people are indeed able to switch between focused and defocused attention [21,23] during the generation of ideas. This notion aligns with the theories on the goal-directedness of creativity [14], according to which the solution of ill-defined problems, such as the alternative uses task (AUT), relies on goal-directed mental processes. Indeed, in the AUT, people are requested to recall uses, generate uses considering specific object features and properties, or disassemble objects and consider uses for object parts [15]. In order to complete all these operations, goal-directed mental processes, such as updating and inhibition, are necessary to monitor the incoming information and suppress proactive inferences [14].

Based on the logic that people with a formal musical background usually show higher competencies in WM [45], it is reasonable to hypothesize that the higher the exposure to a formal music background, the higher the WM competencies, and the higher the DT, given that the latter represents a type of thinking requiring goal-directedness in terms of updating and inhibition [14].

Given these premises, the main research hypothesis was formulated as follows: a formal musical background moderates the association between WM and DT, enhancing the effect of WM on DT.

## 2. Materials and Methods

### 2.1. Participants and Procedure

A convenience sample of 83 healthy young adults (Mean_age_ = 19.64 years; SD_age_ = 0.52 years; 33 females) was recruited as a part of a larger study at the University of L’Aquila, Italy. All the participants were recruited voluntarily: they signed the informed consent and completed a socio-demographic questionnaire assessing age, gender, general health, and formal art education, including music. No subjects reported psychiatric and neurological disorders or drug and alcohol addiction. In addition, 39 participants declared to have a formal background in music in the conservatory. After the socio-demographic questionnaire, the participants were requested to perform the digit span forward test and the alternative uses task. The experiment was conducted at The University of L’Aquila (L’Aquila, Italy) in a quiet room of the Socio-Cognitive Processes in the Life-Span Laboratory. The experiment lasted approximately 25 min, and all the participants participated voluntarily without any reward. The research received ethical approval from the University Ethics Committee in accordance with the Declaration of Helsinki.

### 2.2. Measures

*Working memory.* The digit span forward test (DSFT; ref. [46]) consists of a series of one-digit numbers ranging from 2 to 9. In this simple working memory task, the participants had to repeat a sequence of numbers in the order and reverse order in which the numbers were stated by the experimenter. The span was increased by one digit for each successful trial. The test was terminated when a participant either failed to reproduce the correct sequence for two consecutive strings with the same number of digits or the final nine-digit number sequence was successfully reproduced. The participants receive two points for each successful digit sequence reproduction on the second attempt.

*Musical background.* The participants were requested to indicate if they had a formal background in music, answering a single item “Do you have a formal background in music?”. In addition, the participants were asked to specify how long they had been playing an instrument (M = 4.78 years; SD = 5.50 years).

*Divergent thinking.* The alternative uses task (AUT) from the Torrance test of creative thinking TTCT-Form A [47] requests to find as many alternative uses as possible for carton boxes within 10 min. The technical manual was used to score the alternative uses provided by the participants. A single judge, after completing a 20-hour training on creativity and DT, evaluated alternative uses provided by participants. During the training sessions, the judge was instructed on the definitions of creativity and DT as well as the main theoretical frameworks, including the structure of intellect model [48], which were explained and discussed. In addition, the rater was familiarized with instances of alternative uses previously evaluated by a panel of judges. The judge was requested to practice the evaluation of alternative uses according to the following parameters: (1) Fluency (DT-Fluency), which consists of the number of relevant responses having practical use in a specific context (e.g., a carton box can be used as a doll); (2) Flexibility (DT-Flexibility), that is, the number of categories encompassing the relevant ideas provided by the participants; (3) Originality (DT-Originality), which comprises the sum of weights of statistically frequent or infrequent responses provided by the reference sample. The latter index was scored as follows: 0 points for responses provided by 5% or more of 500 people; 1 point for responses provided by 2–4.99% of 500 people; and 2 points for both responses provided by <2% of 500 people and responses not listed in the technical manual. The three scores of the AUT showed good internal consistency as reported in the technical manual [47]. Following Runco and colleagues’ procedure [49], the three indices of DT were converted into z-scores and summed to obtain a composite index of DT (DT-Total score).

### 2.3. Statistics

Analyses were performed using SPSS Statistics version 24 for Windows (IBM Corporation, Armonk, NY, USA). Descriptive statistics and correlations were carried out preliminarily to detect the main features of the study variables and to check their associations. The moderating effect of a musical background in the association between WM and DT was tested by the PROCESS macro for SPSS Model 1; ref. [50] using 5000 resample of bootstrapped estimates with 95% bias-corrected confidence intervals (CIs). Bootstrapping is a non-parametric approach, which bypasses the issue of non-normality [51] and allows for an accurate test of the mediating and/or moderating effects in small to medium-sized samples, e.g., [50,51,52]. The 95% CIs must not cross zero to satisfy the criteria of moderation [50]. All significance was set to *p* < 0.05.

## 3. Results

### 3.1. Preliminary Analysis

The preliminary normality test revealed that DT-Fluency and DT-Flexibility were normally distributed (Kolmogorov-Smirnov Test: Z_DT-Fluency_ = 0.15, ns; Z_DT-Flexibility_ = 0.07, ns), whereas WM and DT-Originality and musical background (years) were not (Kolmogorov-Smirnov Test: Z_WM_ = 0.00, sig; Z_DT-Originality_ = 0.02, sig; Z_Musical background (years)_ = 0.00, sig). For the standard error means, see Figure 1.

In order to assess the potential differences between the musicians and non-musicians in WM and DT, we performed ANOVA, with the WM and DT total scores as the independent variables and the musicians vs. non-musicians as the between factors. The results showed a significant group main effect only for the DT-total scores: the musicians scored significantly higher than the non-musicians in DT [*F*(1, 81) = 5.25; *p* = 0.02, Partial eta squared = 0.06].

In addition, we performed a correlation analysis, which showed that WM was positively correlated with DT-Fluency (*r* = 0.53, *p* < 0.01), DT-Flexibility (*r* = 0.50, *p* < 0.01), DT-Originality (*r* = 0.25, *p* < 0.01), and DT-Total score (*r* = 0.56, *p* < 0.01). These results confirmed that WM was positively associated with DT. Table 1 reports the preliminary descriptive statistics and correlations among the study variables.

### 3.2. Moderation Analysis

To test the hypothesis that a formal musical background moderates the association between WM and DT, enhancing the effect of WM on DT, we carried out a moderation analysis, entering WM as the independent variable and musical background as the moderator (Figure 2).

Given the high correlation among the DT indices and to ensure parsimony, we included only the DT-Total score as the dependent variable. In addition, age and gender were entered as covariates. The results revealed that years of musical background moderated the association between WM and DT (*B* = 0.14, *SE* = 0.05, *t* = 2.65, *CI* 95% = [0.034, 0.246]) at low (*B* = 1.08, *SE* = 0.39, *t* = 2.74, 95% *CI* [0.295, 1.861]), middle (*B* = 1.75, *SE* = 0.28, *t* = 6.32, 95% *CI* [1.198, 2.302]), and high (*B* = 2.52, *SE* = 0.38, *t* = 6.58, 95% *CI* [1.760, 3.286]) levels (Figure 3 and Table 2).

### 3.3. Post Hoc Power Analysis

In order to test the power obtained from the collected data, a post hoc power analysis was conducted. The model reached the power value of 0.99 and satisfied the recommended cut-off of 0.80 [53], suggesting that the sample of 83 was appropriate for testing the indirect effects of the model hypothesized.

## 4. Discussion

The present study aimed to investigate the involvement of a formal musical background in the association between WM and DT in a sample of 83 healthy young adults. For this reason, we performed a moderation model to analyze the interrelationships among these three competences. Several studies have shown that DT represents a type of thinking that requires goal-directed mental processes, such as WM [4]. The relationship between WM and DT, already well documented in the literature, could benefit from activities, such as musical practice, that share similar underlying mechanisms as WM and DT, e.g., the integration of multisensory stimuli and attentional mechanisms. In this sense, and according to previous research [19], the results of this study confirmed our main research hypothesis, revealing that a musical background moderates the association between WM and DT.

On the one hand, these findings provided further evidence of the notion that DT represents a goal-oriented process that requires the involvement of core executive functions [19,54]. Indeed, a handful of studies indicated that WM is involved in a series of creative activities, including the ability to find alternative uses for common objects [55,56]. For instance, some authors argued that WM, through holding, manipulating, and updating relevant information, increases the likelihood of obtaining divergent, novel, and useful ideas [56]. This evidence has been confirmed by neuroimaging studies in which the ventral and dorsal prefrontal cortex (PFC) are thought to be common neural substrates of WM and DT. Therefore, Vartanian and collaborators [57] hypothesized that the ventral and dorsal parts of the PFC are the likely locations where the transfer effects of WM training occur at DT. Moreover, Vartanian et al. [57] indicated that training WM is an essential intervention strategy that is helpful in promoting fluid intelligence in the service of improving DT.

On the other hand, our findings indicated that a formal musical background facilitates the effect of WM on DT. Interestingly, musicians, differently from individuals with no specific musical training or experience, tend to show better strategies to maintain auditory information within the context of WM, e.g., [58]. Specifically, musicians are more trained to listen, control, monitor, and remember information (i.e., the musical note) as well as learn behavioral inhibition when synchronizing with other musicians [33,59]. It is not a casualty that formal music practice involves several cognitively challenging elements, e.g., long periods of controlled attention and keeping musical passages in WM [33], the same mechanism at the base of DT. Therefore, it is reasonable that these better capacities of musicians can trigger WM, allowing a greater ability to think divergently.

The current research showed some limitations worth mentioning and potential future research directions. First, we enrolled a small sample of young adults in the Italian context, which could undermine the generalizability of the results. Future research should confirm our findings by providing a larger sample from different cultural backgrounds. Second, only WM was considered in the moderating model, providing a partial picture of the impact of executive functioning on DT. Future research should evaluate executive functioning, including other core executive functions as well as the higher-order executive functions [60]. As discussed in the introductory section, executive functions are a set of neurocognitive abilities that underlie awareness, goal-directed thinking, and behavior and include three processes that promote DT: not only WM but also inhibition and cognitive flexibility. To promote DT, it is important to have the ability to override an automatic, dominant response and switch between different modes of thinking, and these are skills that involve inhibition and cognitive flexibility, respectively [16]. In addition, considering only WM does not allow for a comprehensive evaluation of the main factors underpinning the ability to think divergently. To provide a comprehensive evaluation of the moderating role of a musical background, future research should also consider extra-cognitive factors, such as personality traits, cognitive styles, and mental strategies, that could affect DT [61,62,63,64,65]. Third, DT was evaluated by a composite score in line with Runco and colleagues’ procedure [49]. Given that DT represents only one of the main indices of creative potential, future research should consider a more granular approach to creative potential, including other components of creative thinking, such as convergent thinking. Finally, another main limitation of this study concerns the procedure used to measure a formal musical background. We asked the participants about their formal background in music and the duration of their experience playing an instrument. This limitation restricts the research on the impact of music on cognitive skills. Additionally, we did not differentiate the effects of various musical experiences. It would have been interesting to assess other factors, such as the type of musical instrument, the age at which they started playing the instrument, and the nature of their training. Therefore, an interesting future perspective could be to investigate whether there are differences in musical styles (such as classical music, pop, and rock) concerning the interrelationships between a formal musical background, WM, and DT.

The findings yielded by our study may also have implications for positive youth development and education [66]. Specifically, the results suggested that a formal musical background represents a facilitating factor in the association between WM and DT. This evidence opens us to the idea that the introduction of a formal education program in music could be helpful for students to improve DT abilities, which are critical in academic achievement. Awareness of the role of musical activities in fostering important cognitive skills, such as WM and DT, could be crucial for improving instructional practices. In particular, creative thinking as a DT dimension may play a key role in developmental age and education (e.g., [67,68]) and has been documented as a predictor of academic achievement and future workplace performance [69,70]. Creative thinking has also been shown to play a key role in everyday problem-solving [71,72] by promoting autonomous and independent behavior and allowing people to solve problems in their daily lives.

## 5. Conclusions

In conclusion, in the present study, we conducted a moderating analysis to investigate the interrelationships between a formal musical background, WM, and DT. Specifically, we found that a formal musical background moderates the relationship between WM and DT. We hypothesize that musical training trains attentional processes and information-processing abilities so that memory and idea-generation mechanisms are faster and more automatic. Additionally, our results show that the interaction of different individual resources can explain the divergence of thinking in individuals who practice music. This is because a musical background is based on associative strategies and the strategy to focus on the relevant elements when integrating sensory, cognitive, and motor stimuli [32]. The same associative and attentional processes are present when an individual faces a problem and tries to identify a better solution by generating several ideas. Thus, musical practices increase DT due to WM competences, which allow DT to activate associative processes and allocate attention resources. In this way, the present research aims to better understand how music practice might interact with and enhance the capacities of WM and DT.

## Figures and Tables

**Figure 1 brainsci-14-00061-f001:**
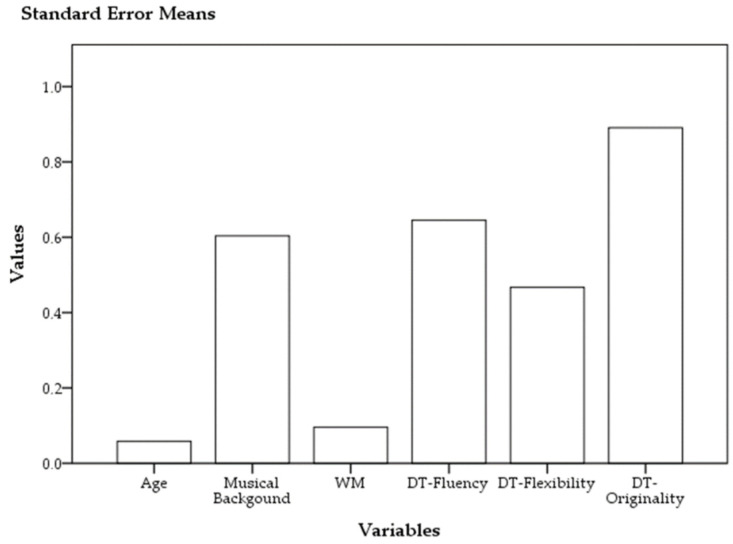
Standard error means.

**Figure 2 brainsci-14-00061-f002:**
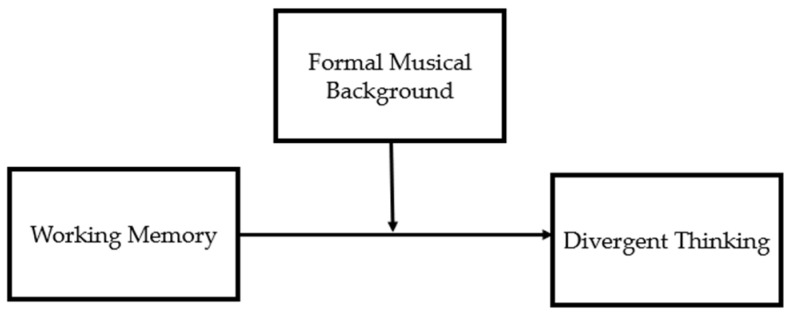
The theoretical moderation model of the current study.

**Figure 3 brainsci-14-00061-f003:**
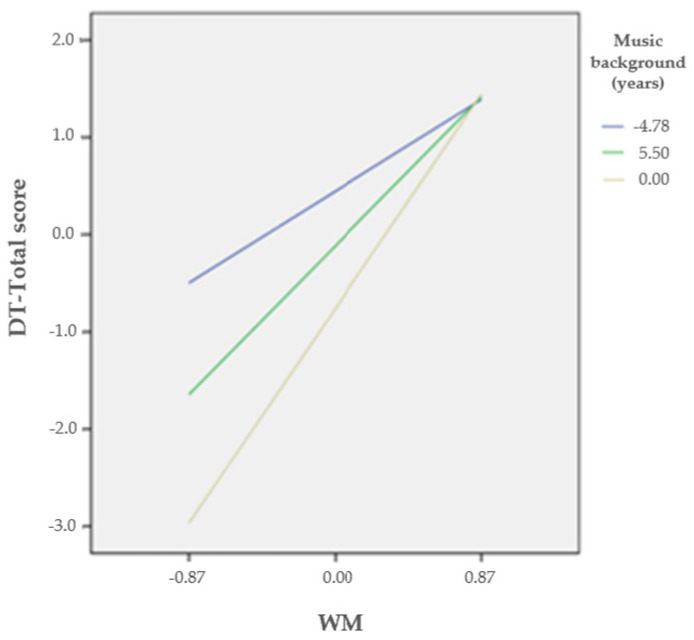
The moderating effect of musical background on the interplay between WM and DT. *Note. N* = 83, WM = working memory, and DT = divergent thinking.

**Table 1 brainsci-14-00061-t001:** Means, standard deviations, and correlations among study variables.

	Mean	S.D.	1.	2.	3.	4.	5.	6.	7.	8.
1. Age	19.64	0.53	1							
2. Gender	0.48	0.50	−0.03	1						
3. Musical background	4.78	5.50	−0.15	−0.05	1					
4. WM	4.46	0.87	−0.10	−0.13	0.15	1				
5. DT-Fluency	12.06	5.88	0.06	−0.15	0.15	0.53 **	1			
6. DT-Flexibility	8.48	4.26	0.13	−0.24 *	0.10	0.50 **	0.89 **	1		
7. DT-Originality	12.22	8.12	0.16	−0.20	0.03	0.25 **	0.22 *	0.34 **	1	
8. DT-Total score	0.00	2.83	0.11	−0.20	0.12	0.56 **	0.96 **	0.95 **	0.43 **	1

*Note. N* = 83, Gender (0 = F; 1 = M) and musical background—yes/no (0 = no; 1 = yes) were dummy coded. WM = working memory; DT = divergent thinking. ** *p* < 0.01 (two-tailed); * *p* < 0.05 (two-tailed).

**Table 2 brainsci-14-00061-t002:** Coefficients for the moderating models.

	B	SE	t	*p*	LLCI	ULCI
WM	1.75	0.28	6.32	0.00	1.198	2.302
Musical background	−0.12	0.12	−0.99	0.15	−0.353	0.119
WM × musical background (interaction)	0.14	0.05	2.65	0.01	0.034	0.246
Age	1.03	0.45	2.29	0.01	0.133	1.930
Gender	−0.77	0.36	−2.13	0.69	−1.482	−0.048
*R*^2^ = 0.48 *F*(6, 76) = 11.73 ***						

*Note. N* = 83. SE = standard error, LLCI = lower limit of the 95% confidence interval, ULCI = upper limit of the 95% confidence interval. *** *p* < 0.001.

## Data Availability

The data presented in this study are available upon request from the corresponding author. The data are not publicly available due to privacy restrictions.

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
