# Peer review of "The Association between Working Memory and Divergent Thinking: The Moderating Role of Formal Musical Background"

_brainsci, 2024, doi:10.3390/brainsci14010061_

Round 1

Reviewer 1 Report (Previous Reviewer 1)

Comments and Suggestions for Authors

The quality of the manuscript has significantly improved after revisions. I have two more minor suggestions.

Although the total DT score was converted into z-scores, I still suggest the authors report the mean of total DT scores in Table 1. Moreover, I also recommend the authors add the mean and standard deviation of gender as they use this variable in correlation analysis.

Besides formal musical background, are there any other forms of educational experience or activities that have the potential to affect working memory and divergent thinking positively? How, then, does this study distinguish itself from other similar studies? It is recommended that the authors strengthen their argument for the significance of the study in the discussion. 

Author Response

The quality of the manuscript has significantly improved after revisions. I have two more minor suggestions.

We thank the reviewer for their comments and suggestions, which have enabled us to improve the work.

Although the total DT score was converted into z-scores, I still suggest the authors report the mean of total DT scores in Table 1. Moreover, I also recommend the authors add the mean and standard deviation of gender as they use this variable in correlation analysis.

In line with the reviewer's suggestion, we have included the scores for both total DT and gender data.

Besides formal musical background, are there any other forms of educational experience or activities that have the potential to affect working memory and divergent thinking positively? How, then, does this study distinguish itself from other similar studies? It is recommended that the authors strengthen their argument for the significance of the study in the discussion. 

The reviewer is right. To date, the literature to date does not definitively state which art form has a greater impact on WM and DT. Musical training has certainly been more extensively studied than other art forms such as painting, sculpture, photography, and dance. However, the available data does not allow us to draw clear conclusions, emphasizing the need for further studies in this field. We addressed the reviewer's suggestion in the discussion session, specifically in lines 359-377

Reviewer 2 Report (Previous Reviewer 2)

Comments and Suggestions for Authors

The manuscript has good flow.

All changes were added in the manuscript

Author Response

We thank the reviewer for their comments and suggestions, which have enabled us to improve the work.

This manuscript is a resubmission of an earlier submission. The following is a list of the peer review reports and author responses from that submission.

Round 1

Reviewer 1 Report

Comments and Suggestions for Authors

This paper investigates an interesting topic: the moderating role of formal musical background in working memory and divergent thinking. Overall, the paper is written well with a clear structure. However, I do have some concerns.

In the Abstract, the authors wrote, “These findings suggest that music buffers the positive effect of high-order cognitive processes, such as WM, on the ability to think divergently.” However, from the results of Figure 2, it appears that musical experience enhances the predictive effect of working memory on divergent thinking. In this light, the statement in the Abstract does not appear to be accurate.

In proposing that the formal musical background has a moderating effect, the inference is simplistic without sufficient superordinate theoretical support. It makes sense if we construct a mediation model that musical training enhances working memory, leading to stronger divergent thinking. Therefore, the authors should strengthen the argument about the moderating effect at the theoretical level.

The sample for this study is not small. However, does it have sufficient statistical power? Please add some information to explain this.

The authors only use one item to measure formal musical background. IS THIS COMMON in this field? In addition, this study did not distinguish the effects of different musical experiences. Although the authors acknowledge this as a limitation, a more profound discussion is needed.

Usually, DT ratings may have some subjective bias and, therefore, require multiple raters to complete this work. In this study, although the authors state that this rater was qualified (completing a 20-hour training on creativity and DT), it is still unclear what was involved in this training. Moreover, even with training, subjective bias cannot be avoided entirely. Therefore, the authors should discuss potential bias caused by using only one rater.

In Table 1, please add the mean of the DT-total score.

Besides music training, many other forms of arts training have the potential to affect working memory and divergent thinking positively. How, then, does this study distinguish itself from other similar studies? It is recommended that the authors strengthen their argument for the significance of the study in the discussion.

Reviewer 2 Report

Comments and Suggestions for Authors

What happens to the study if subjects have no idea about music?

Any specific reasons for selecting 83 subjects ? explain

What happen to the study if subjects are not interested to hear music?

Why did you selected the university subjects? if you try with other subjects what is the status? did you conduct any preliminary studies ?

 Need more explanation about the experimental setup.

How did you selected the subjects?

Compare working memory and divergent  thinking.

Did you apply this study in real time?